# Adaptive Laboratory Evolution and Carbon/Nitrogen Imbalance Promote High-Yield Ammonia Release in *Saccharomyces cerevisiae*

**DOI:** 10.3390/microorganisms13020268

**Published:** 2025-01-25

**Authors:** Alex Pessina, Anna Giancontieri, Tommaso Sassi, Stefano Busti, Marco Vanoni, Luca Brambilla

**Affiliations:** 1Department of Biotechnology and Biosciences, University of Milano-Bicocca, Piazza Della Scienza 2, 20126 Milan, Italy; alex.pessina@unimib.it (A.P.); a.giancontieri1@campus.unimib.it (A.G.); t.sassi@campus.unimib.it (T.S.); stefano.busti1@unimib.it (S.B.); marco.vanoni@unimib.it (M.V.); 2SYSBIO Centre for Systems Biology, 20126 Milan, Italy

**Keywords:** *S. cerevisiae*, adaptive laboratory evolution, bioreactor, protein-based medium, ammonia

## Abstract

Ammonia, essential for fertilizers and energy storage, is mainly produced through the energy-demanding Haber–Bosch process. Microbial production offers a sustainable alternative, but natural yeast cells have not yet demonstrated success. This study aimed to enhance ammonia production in *Saccharomyces cerevisiae* by optimizing amino acid utilization through its deamination metabolism. Adaptive laboratory evolution is a method for rapidly generating desirable phenotypes through metabolic and transcriptional reorganization. We applied it to the efficiently fermenting *S. cerevisiae* strain CEN.PK113-7D using an unbalanced carbon/nitrogen medium to impose selective pressure. We selected several evolved strains with a 3–5-fold increase in amino acid utilization and ammonia secretion. The multi-step bioreactor strategy of the evolved strain AAV6, supplemented with concentrated nitrogen sources, resulted in the production of 1.36 g/L of ammonia, a value in line with levels produced by other microbial systems. This proof-of-concept study suggests that yeast-based processes can be adapted straightforwardly to ammonia production from high-protein waste derived from several sources.

## 1. Introduction

The colorless gas ammonia (NH_3_) is one of the most produced chemicals worldwide, with a global production quantity of 235 million metric tons [1]. About 70% is used in the agricultural sector to synthesize nitrogenous fertilizers [2], with the remainder used for a wide range of applications, such as plastics, pharmaceuticals, explosives, dyes, and other industrial chemicals [3]. Ammonia can also be used as a low-carbon fuel and as energy storage due to its high hydrogen volumetric content (121 kg m^−3^) and weight fraction (17.6%) [1,4]. Most synthetic ammonia is produced nowadays through the Haber–Bosch process, which requires high temperature (400–650 °C) and pressure (100–400 bar) operating conditions [5]. Therefore, this process is highly energy-demanding, accounting for 2% of the global energy supply [6]. Moreover, since it mainly relies on the use of fossil fuels, it is responsible for direct emissions of 450 Mt CO_2_ every year [7]. The shift toward a carbon-free society should involve taking into consideration ammonia production’s impact. Therefore, the future demand for ammonia must be met by seeking different sustainable solutions. To this end, microorganisms’ exploitation represents a practical option thanks to their plasticity. Microbial ammonia production has been achieved by combining various genetic manipulations with several protein-based biomasses. In *Azotobacter vinelandii*, the disruption of various genes led to increased natural ammonia secretion, making this microorganism interesting from an agricultural perspective [8]. Furthermore, ammonia-producing bacteria from different biomasses were achieved by the metabolic engineering of *Bacillus subtilis* and *Escherichia coli* [9,10,11].

Even though the yeast *S. cerevisiae* is one of the most suitable and used microorganisms in biotechnology, ammonia production from wild-type cells has yet to be reported [12]. Indeed, defects in genes related to ammonia uptake or nitrogen compound degradation are associated with unsustainable or low growth in this organism. Watanabe and colleagues accomplished ammonia production outside the cell by expressing a heterologous catabolic enzyme on the yeast surface and feeding cells with single amino acids or mixtures [4,13].

This study uses the vast biological, genetic, and fermentation knowledge accumulated for budding yeast to exploit *S. cerevisiae* deamination metabolism to increase ammonia release through better amino acid utilization. An essential step in our strategy was using an ALE protocol, which is known for quickly generating desirable phenotypes, like stress tolerance or the ability to utilize non-natural substrates, through mutations in metabolic enzymes and the reorganization of transcriptional and metabolic pathways [14]. Thus, we used an unbalanced carbon/nitrogen medium to promote ammonia excretion as a selection medium for ALE, resulting in strains with improved ammonia production and amino acid utilization. Through the multi-step fed-batch fermentation of the evolved strain, we obtained ammonia accumulation in the medium at a level comparable to that obtained by other microbiological systems [15]. Therefore, this study provides strains and fermentation protocols that could be the basis for the successful use of *S. cerevisiae* for sustainable ammonia production from proteinaceous wastes.

## 2. Materials and Methods

### 2.1. Strains and Shake-Flask Cultivation

The *S. cerevisiae* strains used are listed in Table 1. The strains were stored at −80 °C in a YPD medium containing 10 g/L yeast extract, 20 g/L peptone, and 20 g/L D-glucose, with 30% (*v*/*v*) glycerol. Before any experiments were conducted, aliquots of each strain were refreshed on a YP medium (yeast extract 10 g/L and peptone 20 g/L) at 30 °C for two days. For adaptive laboratory evolution and ammonia production, strains were grown in a YPD or YP medium buffered with 0.4 M KH_2_PO_4_ at pH 5.0. Shake-flask cultivations were carried out in 250 mL flasks containing 50 mL of a fresh medium, inoculated with cultures from the logarithmic phase at an optical density (OD) of 0.2 at 660 nm as the initial OD (approximately 0.04 g/L cells, dry weight). The cultures were then incubated at 30 °C with shaking at 160 rpm. Samples (1 mL) were collected to assess growth every 4 h and, upon approaching the stationary phase, every 24 h. For metabolite analysis, an additional 1 mL sample was centrifuged at 16,000× *g* for 3 min at room temperature, and the supernatant was then collected and stored at −20 °C.

### 2.2. Dry Weight Determination

Biomass dry weight was quantified by directly measuring the weight of the dried samples on a precision balance (Denver Instrument SI-234, Bohemia, New York). Approximately 50 OD was centrifuged at 16,000× *g* for 3 min at room temperature, washed three times with deionized water, and dried in a vacuum concentrator (Eppendorf Vacufuge plus, Eppendorf, Hamburg, Germany) at 45 °C in a pre-weighted 1.5 mL Eppendorf tube for ~16 h until a constant weight was reached.

### 2.3. Correlation Between OD660nm and Dry Weight

Shake-flask growth was monitored by determining the development of the cellular optical density over time. We measured the OD at 660 nm, taking two readings for each sample to minimize variability. The OD values were measured after appropriately diluting samples in the same medium to maintain a linear range of 0.1 to 0.6 OD (Ultrospec 500 pro, Amersham biosciences, Amersham, United Kingdom). To establish a correlation between OD and DW (g/L), we performed a regression analysis (Appendix A). The resulting linear equation enabled us to convert OD measurements from subsequent experiments back to DW, facilitating a reliable estimation of biomass concentration. All relationships exhibited a minimum observed R^2^ value greater than 0.98.

### 2.4. Bioreactor Cultivation

Bioreactor cultivation was carried out in a YP medium in an autoclaved 2 L water-jacketed bioreactor (Biostat B, B. Braun, Melsungen, Germany) with a 1.5 L useful volume. A Biostat B control unit monitored temperature, pH, aeration, and foam formation. The temperature was set at 30 °C, with aeration and mixing regulated as a function of dissolved oxygen (DO), which was maintained above 20%. The filtered air was fed through a sparger at a flow rate of 1 VVM. The pH was automatically adjusted to 5 using 2M KOH and 2M H_3_PO_4_. Foam formation was controlled using a silicon polymer suppressor (Antifoam A, Sigma-Aldrich, Darmstadt, Germany). Cells precultured in a shake flask were harvested by centrifugation at 4000× *g* for 5 min, washed twice with cold, sterile water, and resuspended in 5 mL sterile water before being transferred into the bioreactor. OD was measured to inoculate the bioreactor at 0.2. Samples were collected for OD, DW, ammonia, peptides, and amino acids determinations. Before use, the reactors, antifoam bottles, and pH control solution were autoclaved at 121 °C for 30 min.

### 2.5. Three-Step Fermentation

Using the same equipment used for batch cultivation, we carried out a 3-step fermentation strategy with a 1 L working volume. The duration of the batch phase was 24 h. After this period, we initiated a linear feeding strategy using a fresh YP concentrated medium (500 mL, yeast extract 100 g/L, peptone 200 g/L) with a peristaltic pump set at 25 mL/h. When no changes in the optical density were observed, we started the third phase by adding 3% trehalose as a single addition.

### 2.6. Ammonia, Trehalose, Glycerol, and Acetic Acid Determination

The concentration of ammonia in the supernatant was measured using the enzymatic assay K-AMIAR (Ammonia-Rapid kit, Megazyme, Wicklow, Ireland). At the end of the reaction, the absorbance of the mixture at 340 nm was read twice until no further movements were observed. The background ammonia concentrations in different media were subtracted from the final values for each determination. All calculations were performed using the Megazyme Excel calculation sheet (https://prod-docs.megazyme.com/documents/Data_Calculator/K-AMIAR_CALC.xlsx, accessed on 1 December 2024). Trehalose, glycerol, and acetic acid concentrations were determined by chromatographic analysis using a Jasco HPLC system (Jasco Europe, Cremella, Lecco, Italy). The injection volume of the samples and standards was 20 μL. Trehalose, glycerol, and acetic acid were monitored using an Aminex-HPX-87H (Bio-Rad, Hercules, CA, USA) and a cation H+ micro guard cartridge (Bio-Rad Laboratories, Hercules, CA, USA). Analyses were performed isocratically at 0.5 mL/min and 55 °C. The mobile phase consisted of 5 mM H_2_SO_4_, and the compounds were detected using a refractive index detector.

### 2.7. Amino Acid Analysis

Amino acid analyses were performed using a previously described method [19]. The derivatized amino acids were separated using a Waters (Milford, MA, USA) XTerra RP18 Column (4.6 mm × 250 mm i.d, 5 μm particle size) equipped with a precolumn Security Guard (Phenomenex, Torrance, CA, USA). Separation was performed at 40 °C. The flow rate of the mobile phase was 1 mL/min throughout the analysis. The precolumn derivatization reagent o-phthalaldehyde (OPA) was prepared by dissolving 25 mg of OPA and 25 mg of 3-Mercaptopropionic acid in 5 mL of a 0.2 M borate buffer, with a pH of 10.2. The 9-Fluorenylmethyloxycarbonyl (FMOC) stock solution was prepared by dissolving FMOC in acetonitrile (1.25 mg/mL). All the solutions were flushed with nitrogen and stored in the dark at 4 °C until any sign of degradation was observed. The derivatization reaction was initiated by adding 5.5 μL of the derivatization reagents to 5.5 μL of the sample and bringing the quantity up to 200 μL with water. Mobile phases A (40 mM Sodium Phosphate Buffer, pH 7.8) and B (30% acetonitrile; 60% methanol; 10% H_2_O) were used for HPLC separation. Before each sample was processed, the column was equilibrated with 94.5%/5.5% (*v*/*v*) of mobile phase A–B for 10 min. The elution gradient was set as follows: 94.5%/5.5% for 0.85 min; 2.15 min linear gradient to 13% phase B; 23 min linear gradient to 54% phase B; 7 min linear gradient to 66% phase B; and 94.5%/5.5% for 7 min. Uv detection was initially set at 338 nm until min 27 and then switched to 265 nm. A standard amino acid solution was prepared in 0.1 N HCL, and the peaks were identified by comparison with retention times.

### 2.8. Adaptive Laboratory Evolution and Screening

We applied the adaptive laboratory evolution strategy to the strain CEN.PK 113-7D in a YP medium (yeast extract 10 g/L, peptone 20 g/L).

The initial strain was inoculated in 50 mL shake flasks and grown at 30 °C and 160 rpm. Every ten days, 5 mL of the culture was transferred to 50 mL of fresh YP. This process was repeated 10 times, during which evolved clones from different cycles were selected and tested. Simultaneously, samples of the same culture were transferred onto YP medium plates (yeast extract 10 g/L, peptone 20 g/L, and technical agar 20 g/L) to obtain approximately 500 cells per plate. The plates were incubated at 30 °C. After 5–7 days, the largest colonies were selected for more accurate screening. For each cycle of inoculation, the cultures completed four–five duplications. Single clones were screened for growth in 15 mL tubes containing 3 mL of a fresh medium (YP) and were grown at 30 °C. After 72 h, growth was measured at OD. The clones with larger colonies on plates and the best growth were named AAV1 through AAV7 and selected for further experiments.

### 2.9. Total Peptide Determination

The supernatant fraction was determined with 0.1 mL of culture preventively evaporated in a vacufuge (Eppendorf, Hamburg, Germany). The total protein content was determined using the modified Biuret method [20]. Samples were suspended in 0.9 mL KOH 1M and boiled for 10 min. After cooling on ice, 0.1 mL of CuSO_4_ × 5H_2_O 250 mM was added to a final concentration of 25 mM. After 5 min at room temperature, the samples were centrifuged for 2 min, and the absorbance of the supernatant was read at 550 nm. BSA was used as the standard.

### 2.10. Specific Growth Rate, Ammonia Yield, and Nitrogen Consumption Ratio

The maximum specific growth rate, µmax (h^−1^), was calculated by the following equation:$$µ=ln2/t_{D}\;$$
where t_D_ is the time required for duplication measured by optical density (OD). Ammonia yields were calculated based on total nitrogen uptake—grams of ammonia produced divided by the sum of nitrogen grams derived from peptides—using a conversion factor of 6.25 [21] and from individual amino acids [21]. We introduced the nitrogen consumption ratio (TN/DW) to evaluate the amount of nitrogen consumed to produce 1 g of DW. TN/DW was calculated by taking the total amount of nitrogen consumed expressed as the sum of amino acids and peptides (g/L) and dividing it by the maximum DW at 72 h of growth when no more ammonia secretion was observed.

### 2.11. Statistical Analysis

Data were represented as the mean ± SD. Student’s *t*-test and one-way Welch’s ANOVA followed by post hoc tests (Games–Howell, Dunnett) between wild-type strains and between the evolved (AAV) clones with their isogenic parental strain were performed using JASP (a graphical user interface for R: https://jasp-stats.org/, accessed on 1 December 2024). *p* values < 0.05 were considered statistically significant.

## 3. Results

### 3.1. Growth on Carbon/Nitrogen Unbalanced Sources Enhances Ammonia Release in Saccharomyces cerevisiae

YP is a classic rich medium for *S. cerevisiae* which contains a mixture of peptides, amino acids, growth factors, purine and pyrimidine bases, carbohydrates, and water-soluble B-group vitamins [22]. In addition to amino acids and peptides, YP contains trehalose, acetate, and glycerol, which could support some yeast biomass accumulation (Appendix A). YP is usually supplemented with a carbon source, such as glucose (YPD). Given the known interconnection between glucose catabolite repression and amino acid and ammonium transport [23,24], we grew the same *S. cerevisiae* strains in the presence of 2% glucose (YPD) or with no sugar supplementation (YP). All the tested strains successfully grew in YP media, even without the addition of glucose to the medium. The final biomass exhibited significant variation among the tested strains, particularly in YPD (Figure 1A,B), emphasizing the role of glucose as the primary carbon source in sustaining biomass accumulation. Yeast strains grown in the YPD medium until the stationary phase (Figure 1A) did not consume all the available amino acids and proteins/peptides (Figure 1C,D, left bars, and Appendix A). Despite differences in growth kinetics among different YPD-grown strains, the uptake of amino acids and peptides consistently aligned with biomass production as estimated by the obtained DW. Strains grown until the stationary phase in YPD released low ammonia levels in the extracellular environment (Figure 1E, left bars), in agreement with Rojas et al.’s findings [25].

YP-grown cells attained a significantly lower final DW than YPD-grown cells, with lower amino acid and peptide uptake (Figure 1C,D). At the end of the growth phase, we observed the complete consumption of the largest carbon source in YP, trehalose (Appendix A). Additionally, there was a partial uptake of peptides and amino acids (Appendix A), suggesting that certain amino acids and peptides served as carbon sources rather than being exclusively used for protein biosynthesis. Presumably, this scenario resulted in enhanced ammonia production, eventually leading to a 2–5-fold increase in ammonia released in the spent growth medium (Figure 1E). The nitrogen consumption ratio (TN/DW, see the Materials and Methods section), used to estimate the mass-normalized amount of nitrogen consumed by the cells, was, on average, higher in YP than in YPD, with statistical significance for CBS8272, CBS8267, and CEN.PK113.7D (Figure 1F).

### 3.2. Increased Amino Acid Catabolism Correlates with High Ammonia Secretion

Yeast cells can synthesize amino acids de novo but also actively import them from the medium [26]. Enhanced amino acid metabolism provides an energetic advantage, and amino acid breakdown provides carbon and nitrogen ready for use. We applied adaptive laboratory evolution (ALE) to the representative *S. cerevisiae* CEN.PK113-7D strain to test the possibility of isolating yeast variants with better fitness on YP (Figure 2A). We chose this strain based on its well-documented advantages, including its efficient fermentation [27,28], genetic manipulability (its isogenic diploid progenitor CEN.PK2 has been used for functional genomics studies [29], yielding about 2000 deletions available at EUROSCARF—European *Saccharomyces cerevisiae* archive for functional analysis), extensive use in biotechnological applications and productions [30], and rich history as a laboratory standard [31]. We isolated seven CEN.PK113-7D-derived clones, which we designated as AAV1-7. Strains AAV2-7 showed no decrease in overall growth rate compared to the non-evolved wild-type strain (0.28 h^−1^) and can complete one further round of mass duplication. The only exception was the evolved strain AAV1, showing a growth rate of 0.19 h^−1^ and reaching a lower final DW (Figure 2B). All the evolved strains consumed, on average, about 3–4-fold more amino acids than their wild-type isogenic counterparts (Figure 2C), while no significant difference was observed in the peptide uptake rate (Figure 2D). Ammonia production increased by three to five times (Figure 2E), reflecting the higher usage of amino acids. The uptake of seven proteinogenic amino acids (Asp, Glu, Asn, Ser, Arg, Ala, and Trp) was enhanced (Appendix A). Once again, we observed a trend of increased nitrogen uptake per DW unit, with statistically significant results for clones AAV1, AAV5, AAV6, and AAV7 (Figure 2F). Because of its slow growth phenotype, strain AAV1 was not considered further, and the evolved strain AAV6 was used in the following bioreactor experiments.

### 3.3. Improved Amino Acid Uptake Yields Significant Ammonia Production

The bioreactor batch fermentation of the AAV6 strain showed a remarkable increase in biomass yield resulting from a reduction in total amino acid and peptide uptake (Figure 3B). However, this approach did not improve the NH3 production process (Figure 3A). Therefore, we transitioned to a three-step fermentation strategy to further stimulate amino acid uptake and, ultimately, ammonia accumulation. Initially, the AAV6 strain was cultured in batch mode until it reached a DW of 0,17 g/L (Figure 3C). After 24h, 500 mL of a fresh, concentrated medium (yeast extract 100 g/L–peptone 200 g/L) was provided in feeding mode for 20 h, resulting in a DW of 5 g/L and 0.53 g/L of ammonia produced (Figure 3D). At the end of the second phase, a significant amount of residual amino acids remained in the medium. In particular, the amino acids glutamate and aspartate, which tended to be absent in the batch bioreactor tests (Appendix A), were still present (Appendix A). At this point, a third phase was initiated by adding 30 g/L trehalose to allow the cells to continue consuming amino acids and avoid ammonia uptake. The uptake of residual amino acids was observed (Appendix A), and the DW of the culture increased to 19,6 g/L. This protocol allowed us to obtain 1.36 g/L of ammonia (Figure 3D) with amino acid and peptide uptake values of 8.8 g/L and 10.1 g/L, respectively. The final ammonia yield relative to the whole amino acid and peptide usage was 0.34 g/g.

## 4. Discussion

Many studies have reported ammonia accumulation in bacteria grown in protein-rich media [13,32,33]. In this condition, amino acid deamination produces ammonia molecules that are eventually released outside the cells [10]. Most attempts have been directed toward using *B. subtilis* or *E. coli* harboring deletions of genes involved in nitrogen metabolism [9,10,11,15]. Yeast has generally not been considered for two main reasons: firstly, the deletion of genes involved in nitrogen metabolism is deleterious and leads to poor growth [34,35,36]; moreover, ammonia is a preferred nitrogen source for yeast cells, which are more inclined to absorb ammonia from the environment rather than to release it [37]. However, low levels of extracellular ammonia in the presence of abundant nitrogen compounds are also occasionally observed [38,39].To date, successful ammonia production from yeast has been achieved only by using strains that express either glutaminase or L-amino acid oxidase on the cellular surface. This approach enhances enzyme efficiency by allowing direct interaction with extracellular substrates [4,13]. In this study, we demonstrate the possibility of using the yeast *S. cerevisiae* by exploiting its metabolism to produce ammonia naturally. Using the adaptive laboratory evolution of the wild-type strain CEN.PK113-7D in a YP medium (i.e., in a condition promoting ammonia secretion in a shake flask, Figure 2A), we isolated different clones characterized by strong biomass formation, enhanced amino acid usage, and impressive ammonia production compared to the wild-type strain (Figure 2B,C,E). As shown in Figure 2C and in comparison with Figure 2D, the improved release of ammonium may be associated with increased amino acid uptake and consumption. Besides aspartate and glutamate, the evolved strains used other amino acids such as asparagine, serine, arginine, alanine, and tryptophan, contributing to their phenotype of overall increased amino acid uptake (Appendix A). When strains were grown in the presence of glucose, the phenotypical changes acquired by ALE disappeared, and little ammonium was generated (Appendix A).

In both wild-type and evolved strains, as well as in shake flasks or a fermenter, growth in the glucose-free medium is mandatory to stimulate ammonia production. This feature is similar to what has been observed in *E. coli* [11]. Previous studies in yeast suggested that a higher consumption of amino acids is associated with a catabolite derepressed state, a process involving the PKA and TOR pathways [24]. Once internalized, the amino acid’s carbon skeleton is directly used as a carbon source [40], while ammonia resulting from deamination can be stored as glutamic acid, glutamine, and branched-chain amino acids [41]. We speculated that ammonia excretion in YP is due to excess nitrogen not being employed for biomass production [42]. The differences in the concentrations of produced ammonia between YPD and YP could be related to the main carbon source forming the biomass. Since nitrogen accounts for 9% of yeast biomass [20,43] while constituting about 16% of average proteins [21], the growth on amino acids results in a surplus of nitrogen. Due to its toxic effect and in accordance with previous observations on nutrient sources that function as both carbon and nitrogen [42], the share of nitrogen not necessary for biomass formation would be released outside the cells. Indeed, in the batch fermentation of the evolved strain, nearly two-thirds of input nitrogen present in amino acids and peptides (Figure 4, blue and red portions of the nitrogen input bar) does not accumulate in the biomass but instead is excreted as ammonia in the medium (Figure 4, green portion of the nitrogen output bar).

Given the central role of amino acid transport in ammonia excretion, a genetic and transcriptional analysis of the amino acid transporters expressed during fermentation may identify the specific transporter involved. This analysis could suggest strategies to enhance amino acid transport, possibly leading to improved ammonia release. Our results show that *S. cerevisiae* can release ammonia from nitrogen-enriched substrates in response to an imbalance in the C/N ratio. Therefore, contrary to what has been reported in previous studies [13], yeast can be used to produce ammonia through its metabolism, which is in line with previous findings in other microorganisms [10,11,15,33]. Yeast showed specificity in amino acid utilization, and the ALE protocol expanded the range of employed substrates. Growing the evolved AAV6 strain according to a three-step fermentation strategy involving a batch step, a linear feeding strategy with a fresh YP concentrated medium, and a final addition of trehalose, we obtained the release of a total of 1.36 g of ammonia per liter. We selected trehalose as a carbon source because HPLC analysis (Appendix A) showed that it is the most abundant sugar in the YP medium, likely originating from the yeast extract component. Moreover, yeast readily utilizes trehalose without causing catabolite repression. Since ammonia production was hindered by glucose in the YPD medium (Figure 1E), we hypothesized that trehalose would prevent ammonia reuptake and promote its continuous secretion from amino acid uptake. To the best of our knowledge, this represents the highest level of natural ammonia secretion from *S. cerevisiae* observed in the context of intracellular production, closely aligning with typical levels recorded in other microbial systems [12]. We believe that ammonia production can be further enhanced by increasing and maximizing the range of amino acid usage while maintaining cells in a state of catabolite derepression. Although this study focused on producing ammonia using conventional substrates, future improvements in this process may involve protein-rich waste derived from several sources.

## Figures and Tables

**Figure 1 microorganisms-13-00268-f001:**
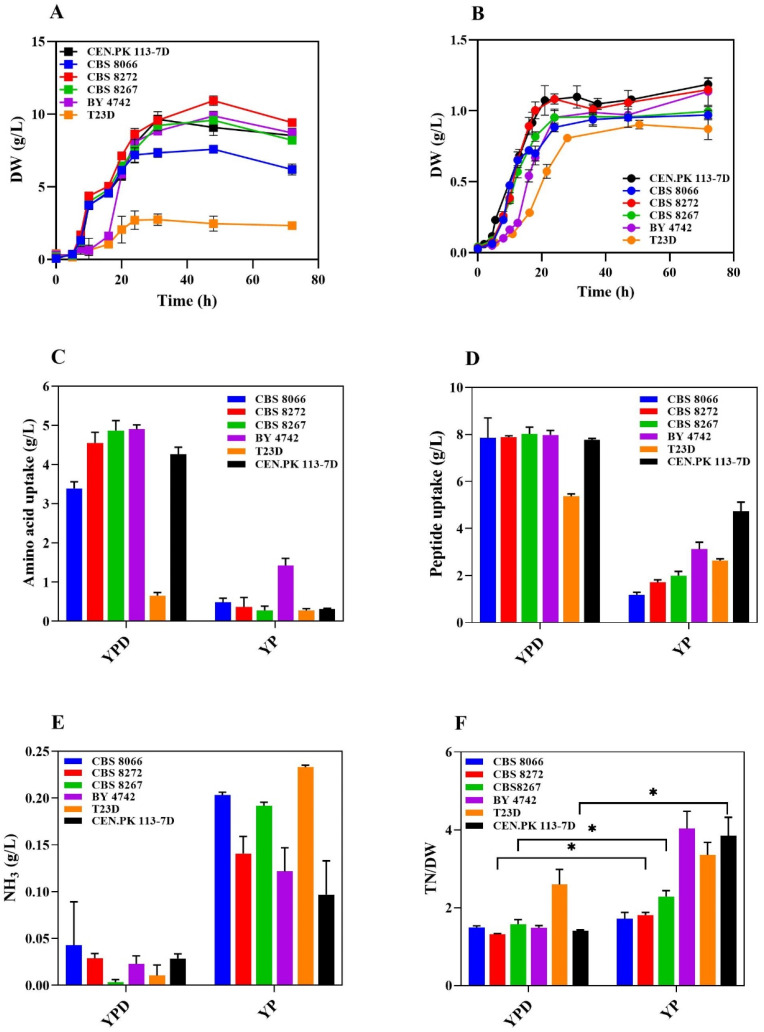
*S. cerevisiae* characterization in YPD and YP. (**A**) Growth curves of different *S. cerevisiae* strains in YPD and (**B**) YP. (**C**) Amino acid uptake, (**D**) total peptide uptake, and (**E**) ammonia production after 72 h. (**F**) Nitrogen consumption ratio (TN/DW) of strains grown in YPD and YP. DW represents the highest value achieved during growth, while TN is the total amount of amino acids and peptides taken up by yeast after 72 h. A high TN/DW ratio indicates high nitrogen consumption per gram of DW. The asterisk (*) indicates significant differences (*p* < 0.05) in TN/DW between the two media for the same strain, obtained using one-way Welch’s ANOVA followed by Games–Howell post hoc tests.

**Figure 2 microorganisms-13-00268-f002:**
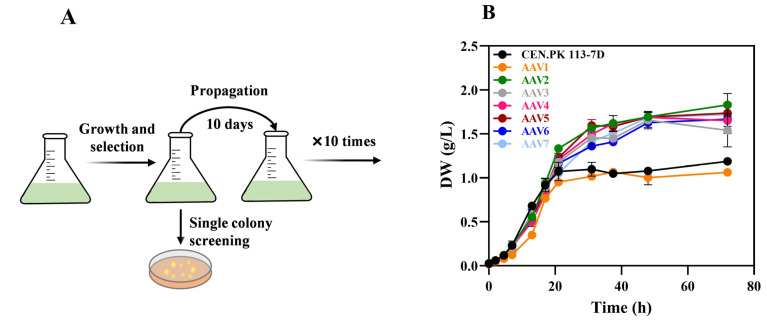
*S. cerevisiae* CEN.PK113-7D and evolved clone characterization. (**A**) This schematic representation illustrates the ALE process used to evolve the CEN-PK113-7D strain successfully, allowing it to utilize a broader range of amino acids. (**B**) The evolved strain (AAV)’s growth curve compared with the WT strain in a YP buffered medium at pH 5. (**C**) Overall amino acid and (**D**) peptide uptake in the medium. (**E**) Ammonia production after 72 h. (**F**) Nitrogen consumption ratio between the strain CEN.PK113-7D and AAV clones in YP. A high TN/DW ratio indicates more nitrogen consumed per gram of DW. The asterisk (*) indicates significant differences (*p* < 0.05), while “ns” indicates no significant differences compared to the control (CEN.PK113-7D) using Student’s *t*-test or one-way Welch’s ANOVA followed by post hoc tests.

**Figure 3 microorganisms-13-00268-f003:**
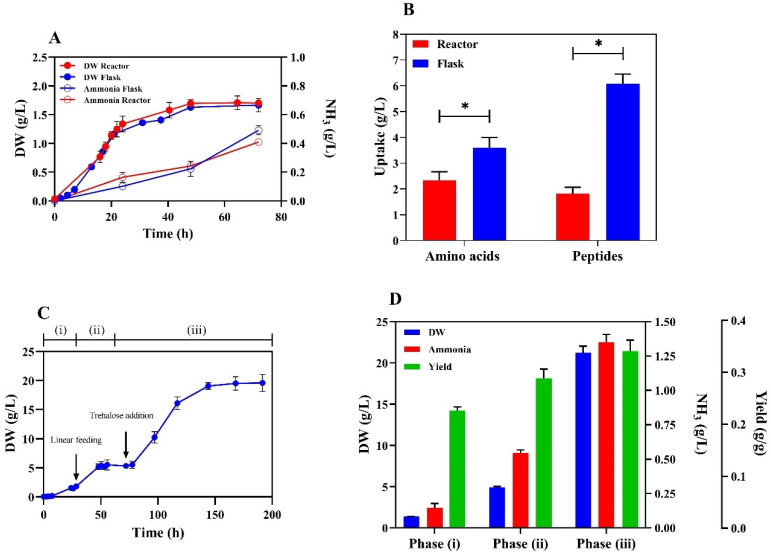
Bioreactor and flask characterization of the evolved strain AAV6. (**A**) Flask and batch bioreactor experiments in YP medium at pH 5. **(B**) Amino acid and peptide uptake at the end of growth in flasks and batch reactor (72 h). (**C**) Growth curve in a 3-step fermentation mode. The experiment comprised three steps: (i) batch cultivation in YP medium at pH 5, (ii) linear feeding of fresh YP concentrated medium (yeast extract 100 g/L, peptone 200 g/L), and (iii) addition of 3% trehalose. (**D**) Calculated parameters during the three-step fermentation strategy with the evolved strain AAV6. DW, ammonia, and yield represent the values achieved at the end of each phase. The yield is the amount of ammonia produced divided by the nitrogen uptake from amino acids and peptides (see M&M). The asterisk (*) indicates significant differences (*p* < 0.05) using Student’s *t*-test.

**Figure 4 microorganisms-13-00268-f004:**
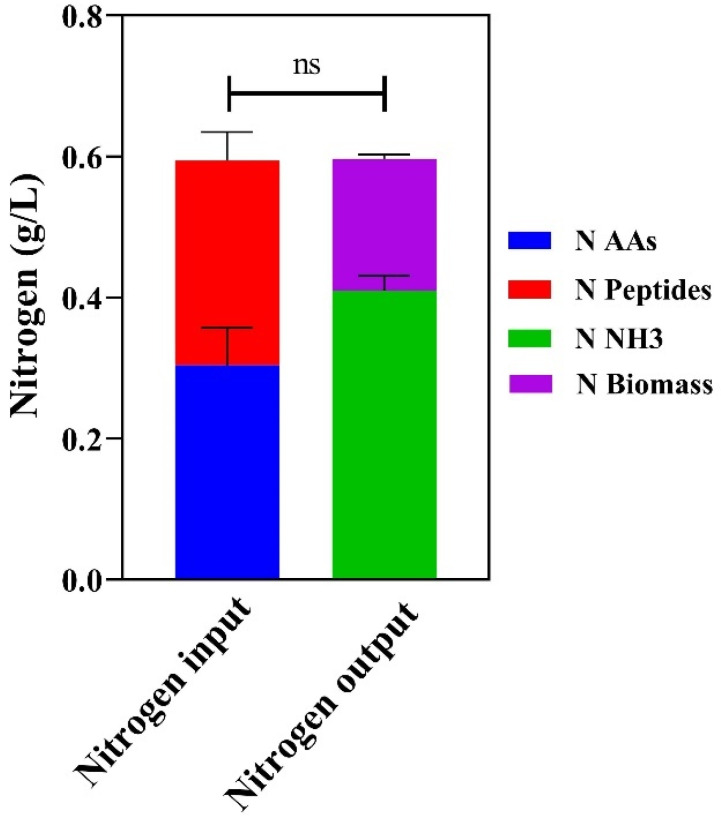
Total nitrogen balance for AAV6 in a batch bioreactor: nitrogen input refers to the total nitrogen usage from amino acids and peptides, whereas nitrogen output indicates the amount of nitrogen contributing to both biomass formation and ammonia production after 72 h of growth in YP. “ns” indicates no significant differences between nitrogen input and output using Student’s *t*-test.

**Table 1 microorganisms-13-00268-t001:** A list of the *S. cerevisiae* strains. Alias: CBS 8272-CEN.PK122; CBS 8267-BAY17. The seven clones selected through ALE are named AAV1 through AAV7.

*Saccharomyces cerevisiae* Strain	Description	Source
CEN.PK113-7D	*MATa MAL2-8c SUC2*	[16]
CBS 8066	*MATa/α HO/ho*	[16]
CBS 8272	*MATa/α*, prototrophic	[16]
CBS 8267	*MATa/α*, prototrophic	[16]
T23D	Meiotic progeny of CBS 8066	[17]
BY4742	*MATα*; *his3Δ1*; *leu2Δ0*; *lys2Δ0*; *ura3Δ0*	[18]
AAV1-7	CEN.PK113-7D clones selected by ALE	This study

## Data Availability

The original contributions presented in this study are included in the article/Appendix A. Further inquiries can be directed to the corresponding author.

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
