# Peer review of "Adaptive Laboratory Evolution and Carbon/Nitrogen Imbalance Promote High-Yield Ammonia Release in Saccharomyces cerevisiae"

_microorganisms, 2025, doi:10.3390/microorganisms13020268_

Round 1

Reviewer 1 Report

Comments and Suggestions for Authors

Review

for the article entitled “Adaptive Laboratory Evolution and carbon/nitrogen imbalance promote high-yield ammonia release in Saccharomyces cerevisiae

The authors employed an adaptive laboratory evolution (ALE) strategy to develop new strains of the yeast Saccharomyces cerevisiae with the objective of increasing ammonia production and amino acid utilisation. The selection medium utilised was an unbalanced carbon/nitrogen medium. The multistep fermentation of the developed strain AAV6, evolved from strain CEN.PK 113-7D, resulted in a 5-fold increase in ammonia release (up to 1.36 g/L), a value comparable to that produced by other known microorganisms. The article is consistent with the scope of the journal "Microorganisms" and provides a novel approach to microbial ammonia production and therefore merits publication in "Microorganisms". However, the authors are encouraged to revise the manuscript.

Point 1: It is essential to emphasize in the abstract and subsequently in the text that increased ammonia release was obtained by strain AAV6 evolved from strain CEN.PK 113-7D.

Point 2: The article should provide a more detailed characterization of the original strain CEN.PK 113-7D, including how it was obtained and its features, rather than simply providing a reference.

Point 3: The “Results” section should be restricted to the results obtained from the present study, without discussion of other investigations. Text on page 5, lines 177-187, which compares the results with those in the literature, should be moved to the “Discussion” section.

Point 4: Repetitions from the Introduction section should be removed, for example lines 277-282.

Point 5: It is very important to position Figure 4 immediately subsequent to its initial mention in the text.

Author Response

Thank you very much for taking the time to review this manuscript and for your helpful comments. Here is our response.

Point 1: It is essential to emphasize in the abstract and subsequently in the text that increased ammonia release was obtained by strain AAV6 evolved from strain CEN.PK 113-7D.

We thank the reviewer for the comment. We have changed the sentences on page 1, Line 18, by adding the indication of the strain that showed increased ammonia release: “We selected several evolved strains with a 3-5-fold increase in amino acid utilization and ammonia secretion. The multi-step bioreactor strategy of the evolved strain AAV6, supplemented with concentrated nitrogen sources, resulted in the production of 1.36 g/L of ammonia, a value in line with levels produced by other microbial systems. This proof-of-concept study suggests that yeast-based processes can be adapted straightforwardly to ammonia production from high-protein-waste derived from several source”. 
Moreover, we emphasize the use of evolved strain AAV6 derived from CEN.PK 113.7D (line 259) and the use of AAV6 in the subsequent bioreactor experiments by changing the initial statement at lines 272-276 as follows: “Bioreactor batch fermentation of the AAV6 strain showed a remarkable increase in biomass yield resulting from a reduction in total amino acid and peptide uptake (Figure 3B). However, this approach did not improve the NH3 production process (Figure 3A) Therefore, we transitioned to a three-step fermentation strategy to further stimulate amino acid uptake and, ultimately, ammonia accumulation. Initially, AAV6 strain was cultured in batch mode until it reached a 10 optical density (Figure 3C).”

Point 2: The article should provide a more detailed characterization of the original strain CEN.PK 113-7D, including how it was obtained and its features, rather than simply providing a reference.

We thank the reviewer for criticizing this point.
The yeast strain CEN.PK113-7D is a well-established laboratory standard known for its high crossing and sporulation efficiency, spore viability, fermentation performance, ease of manipulation, and superior production capabilities (Nijkamp et al., 2012). From PubMed more than one hundred studies have utilized the CEN.PK yeast family for various physiological and metabolic investigations, yielding biotechnological products such as lactate (van Maris AJ et al., 2004), pyruvate, and ethanol (Pagliardini J et al., 2010). Additionally, it serves as a cell factory for numerous other compounds (Zelle RM et al., 2010; Wattanachaisaereekul S et al., 2008; Madsen KM et al., 2011). The strain exhibits efficient growth under both aerobic and anaerobic conditions, with excellent performance in bioreactors, making it ideal for production (Paciello L et al., 2014). 
Although we achieved good results in ammonia production and a favorable TN/OD ratio with other strains (Figure 2E-F), We have reasoned that the extensive utilization of CEN.PK113-7D significantly simplified the strain selection and experimental design in our manuscript making our approach interesting for many groups.
To provide more details about the strain CEN.PK113-7D, we have added the following sentence in the abstract: “We applied it to the efficiently fermenting S. cerevisiae strain CEN.PK113-7D using an unbalanced carbon/nitrogen medium to impose selective pressure.” (Line 16). Moreover, we have integrated the manuscript with a detailed description of the CEN.PK113-7D and references as support: “We applied Adaptive Laboratory Evolution (ALE) to the representative S. cerevisiae CEN.PK113-7D strain to test the possibility of isolating yeast variants with better fitness on YP (Figure 2A). We chose this strain based on its well-documented advantages, including efficient fermentation [27,28], genetic manipulability (its isogenic diploid progenitor CEN.PK2 has been used for functional genomics studies [29], yielding about 2,000 deletions available at EUROSCARF - European Saccharomyces cerevisiae archive for functional analysis), extensive use in biotechnological applications and productions [30], and a rich history as a laboratory standard [31].”(Line 240-247)

Point 3: The “Results” section should be restricted to the results obtained from the present study, without discussion of other investigations. Text on page 5, lines 177-187, which compares the results with those in the literature, should be moved to the “Discussion” section.

We have removed the sentences “Many studies have reported ammonia accumulation in bacteria grown in protein-rich media, in which amino acid deamination produces ammonia molecules eventually released outside the cell. however, low levels of extracellular ammonia in the presence of abundant nitrogen compounds are frequently observed” (lines 193-197). The sentences were moved to the discussion section with minor modifications (lines 305-307, 313-314).

Point 4: Repetitions from the Introduction section should be removed, for example lines 277-282.

We thank the reviewer for the suggestion. We have removed the redundant sentences in the discussion section “Due to massive requests for fertilization and transport applications, worldwide ammonia demand is expected to increase in the coming decades. Ammonia production is primarily carried out via the Haber-Bosch synthesis, and much effort has been devoted in the last ten years to investigate the potential use of microorganisms for sustainable production” (line 301-305). Now, the discussion focuses more on the biological aspects, comparing our findings with those in the literature.

Point 5: It is very important to position Figure 4 immediately subsequent to its initial mention in the text.

We have moved the Figure 4 as suggested.

Reviewer 2 Report

Comments and Suggestions for Authors

The manuscript entitiled ‘Adaptive Laboratory Evolution and carbon/nitrogen imbalance promote high-yield ammonia release in Saccharomyces cerevisiae’ uses the vast biological, genetic, and fermentation knowledge accumulated for budding yeast to exploit the S. cerevisiae deamination metabolism to increase ammonia release through better amino acid utilization. This study provides strains and fermentation protocols that could be the basis for successfully using S. cerevisiae for sustainable ammonia production from proteinaceous wastes. The work is meaningful, but some things need to be improved.

1.     In line 79, the unit of the centrifuged should be revised. I don’t understand the ‘13.000 RPM.’

2.     In line 81, the author should add how long the samples are kept in the drying process and what experimental equipment was used to measure the constant weight and drying.

3.     Some paragraphs are too long. It is very hard to read, given the example of 3.1 and discussion.

4.     What do the AAV samples mean from 1 to 7? Please add the description to the manuscript.

Author Response

Thank you very much for taking the time to review this manuscript and for your helpful comments. Here is our response

Point 1. In line 79, the unit of the centrifuged should be revised. I don’t understand the ‘13.000 RPM.’

We thank the reviewer for the correction. We have changed “rpm” to “xg” alongside the manuscript for the centrifugation steps (lines 85, 92 and 104).

Point 2. In line 81, the author should add how long the samples are kept in the drying process and what experimental equipment was used to measure the constant weight and drying.

We have detailed the protocol by specifying the instruments used and the time required to achieve a constant weight (lines 81 and 94).

Point 3.  Some paragraphs are too long. It is very hard to read, given the example of 3.1 and discussion.

We thank the reviewer for this suggestion. We have splitted long sentences throughout the manuscript. Here some examples:

1)    At the end of the growth phase, we observed complete consumption of the largest carbon source in YP, trehalose (Table S2). Additionally, there was partial uptake of peptides and amino acids (Table S3), suggesting that certain amino acids and peptides served as a carbon source rather than being exclusively used for protein biosynthesis
2)     Many studies have reported ammonia accumulation in bacteria grown in protein-rich media [13,21,22]. In this condition, in which amino acid deamination produces ammonia molecules eventually released outside the cells [10]. 
3)    To date, successful ammonia production from yeast has been achieved only by using strains that express either glutaminase or L-amino acid oxidase on the cellular surface. This approach enhances enzyme efficiency by allowing direct interaction with extracellular substrates.
4)    Given the central role of amino acid transport in ammonia excretion, a genetic and transcriptional analysis of the amino acid transporters expressed during fermentation may identify the specific transporter involved. This analysis could suggest strategies to enhance amino acid transport, possibly leading to improved ammonia release.

Point 4. What do the AAV samples mean from 1 to 7? Please add the description to the manuscript.

We thank the reviewer for criticizing this aspect. We have better clarified the AAV strains by modifying table 1 as follows: “Table 1. List of the S. cerevisiae strains. and mutants. selected by ALE. Alias: CBS 8272-CEN.PK122; CBS 8267-BAY17. The seven clones selected through ALE were named AAV1 through AAV7.” (line 88). We have added detailed information about the screening procedure in paragraph “2.7. Adaptive Laboratory Evolution and Screening” (line 149) and specified the names we assigned to the clones selected (line 162). We have also changed the sentence on page 1, Line 20, by adding the indication of the strain used in bioreactor experiments. Moreover, we have emphasized the use of evolved strain AAV6 derived from CEN.PK 113.7D (line 259) and the use of AAV6 in the subsequent bioreactor experiments by changing the initial statement at line 272-276) as follows: “Bioreactor batch fermentation of the AAV6 strain showed a remarkable increase in biomass yield resulting from a reduction in total amino acid and peptide uptake (Figure 3B). However, this approach did not improve the NH3 production process (Figure 3A) Therefore, we transitioned to a three-step fermentation strategy to further stimulate amino acid uptake and, ultimately, ammonia accumulation. Initially, AAV6 strain was cultured in batch mode until it reached a 10 optical density (Figure 3C).”

Reviewer 3 Report

Comments and Suggestions for Authors

GENERAL COMMENTS

To evaluate ammonia secretion, the authors investigated the growth of Saccharomyces cerevisiae in a carbon/nitrogen unbalanced rich-protein medium. They concluded that this proof-of-concept study suggests that yeast-based processes can be adapted straightforwardly to ammonia production from high-protein-waste derived from several sources.

The manuscript shows an adequate experimental strategy. The results are well-presented and reasonably well-discussed. Nonetheless, the authors must attend to some concerns before it can be accepted for publication.

1. The abstract's introduction is too long (Lines 10-16). Before concluding the abstract, it is recommended that the methodological strategy and the main results be summarized better.

2. Adaptive Laboratory Evolution is an issue missing in the introduction section. The authors must consider including a brief description of this subject.

3. A critical issue is the measurement of the optical density. It is well-known that a measurement greater than 3.0 is not recommended for reliable quantitative measurements. So, when authors report optical density values greater than 3 (even up to 110), the question arises: how is this possible? Did the authors determine the instrument's linearity range of cell density versus optical density? If so, what was the result? Many results and decisions were based on this measurement. The authors must clear this aspect of their work before considering an acceptance of the manuscript.

4. Why did the authors use trehalose as the carbon source during the last phase of fed-batch culture? It must be discussed.

COMMENTS ON ENGLISH

The English is understandable and good, but it could be improved. Some sentences could be wordy, and there are some cases where it would be advisable to split a long sentence into two short sentences to clarify the text. It is recommended that authors carefully review the English throughout the manuscript.

SPECIFIC COMMENTS

1. Line 53. Please consider the sentence "This study uses the vast biological..." to be the beginning of a new paragraph.

2. Line 65-81. Please provide the manufacturer and model of the spectrophotometer used to measure OD.

3. Line 72. Was 0.2 OD the initial OD of the culture? If so, please state it clearly.

4. Line 74. Please provide the volume and frequency of the sample collection.

5. Line 79. Centrifugation speed never must be expressed in "rpm" because it changes according to the diameter of the rotor used. Please use "×g" rather than "rpm" and review it alongside the manuscript.

6. Line 101. Was trehalose incorporated gradually or in a single addition? Please clarify.

7. Line 108-109. Please do not mention amino acids since this paragraph describes the determination of ammonia, trehalose, glycerol, and acetic acid.

8. Line 144-147. Was this screening made at the end of the 10 cycles of inoculation?

9. Line 201. Please provide these data as a supplementary table or figure. The journal's instruction for authors states that "data not shown" should be avoided. 

10. Lines 225 and 227. Did the authors express the growth rate in units µ-1? What does it mean? Please review.

11. Authors use periods or commas as decimal separators interchangeably throughout the manuscript. Please homogenize the use of the decimal separator.

Author Response

We thank the reviewer for taking the time to review this manuscript and for your helpful comments. Here is our response

GENERAL COMMENTS

1.The abstract's introduction is too long (Lines 10-16). Before concluding the abstract, it is recommended that the methodological strategy and the main results be summarized better.

We thank the reviewer for the suggestion. We have modified the abstract by reducing the introduction information and summarizing the results better. Here below, the modified version:

“Ammonia, essential for fertilizers and energy storage, is mainly produced through the energy-demanding Haber-Bosch process. Microbial production offers a sustainable alternative, but natural yeast cells have not yet demonstrated success. This study aimed to enhance ammonia production in Saccharomyces cerevisiae by optimizing amino acid utilization through its deamination metabolism. Adaptive laboratory evolution is a method for rapidly generating desirable phenotypes through metabolic and transcriptional reorganization. We applied it to the efficiently fermenting S. cerevisiae strain CEN.PK113-7D using an unbalanced carbon/nitrogen medium to impose selective pressure. We selected several evolved strains with a 3-5-fold increase in amino acid utilization and ammonia secretion. The multi-step bioreactor strategy of the evolved strain AAV6, supplemented with concentrated nitrogen sources, resulted in the production of 1.36 g/L of ammonia, a value in line with levels produced by other microbial systems. This proof-of-concept study suggests that yeast-based processes can be adapted straightforwardly to ammonia production from high-protein-waste derived from several sources.”

2.    Adaptive Laboratory Evolution is an issue missing in the introduction section. The authors must consider including a brief description of this subject.

We thank the reviewer for pointing out this issue. 
We have added an essential explanation of Adaptive Laboratory Evolution and benefits as follows: “An essential step in our strategy was using an ALE protocol, which is known for quickly generating desirable phenotypes like stress tolerance or the ability to utilize non-natural substrates, through mutations in metabolic enzymes and reorganization of transcriptional and metabolic pathways [14]. Thus, we used an unbalanced carbon/nitrogen medium to promote ammonia excretion as a selection medium for ALE, resulting in strains with improved ammonia production and amino acid utilization.” (line 58-64).
We have added detailed information about the screening procedure in paragraph 2.7. Adaptive Laboratory Evolution and Screening (line 150-154): ” We applied Adaptive Laboratory Evolution strategy in a YP medium (yeast extract 10 g/L, peptone 20 g/L) to the strain CEN.PK 113-7D.
The initial strain was inoculated in 50 mL shake flasks grown at 30°C and 160 rpm. Every ten days, 5 mL of the culture was transferred to 50 mL fresh YP. This process was repeated 10 times, during which evolved clones from different cycles were selected and tested.” 
We have also specified the names we assigned to the clones selected with this procedure by adding the sentence: “Clones with the larger colonies on plates and the best growth were named AAV1 through AAV7 and selected for further experiments “(line 162).

3.    A critical issue is the measurement of the optical density. It is well-known that a measurement greater than 3.0 is not recommended for reliable quantitative measurements. So, when authors report optical density values greater than 3 (even up to 110), the question arises: how is this possible? Did the authors determine the instrument's linearity range of cell density versus optical density? If so, what was the result? Many results and decisions were based on this measurement. The authors must clear this aspect of their work before considering an acceptance of the manuscript.

We thank the reviewer for highlighting this critical point. The linearity of our instrument (Ultrospec 500 Pro, Amersham Biosciences) for turbidity measurements (OD660 nm) was previously verified. The photometric scale was found to be linear between 0.1 and 0.6. Consequently, all measurements were taken within this range relative to the blank. Whenever the range was exceeded, samples were carefully diluted in the same medium or buffer, as well as the blank samples. For example, in the fed-batch experiment at an OD of 110, we diluted the sample 200-fold using serial dilutions with the same fermentation medium. This procedure has been further detailed in the 2.1 Strains and Shake Flask Cultivation" section of the Materials and Methods with this updated paragraph: “Shake flask cultivations were carried out in 250 mL flasks containing 50 mL of fresh medium, inoculated with cultures from the logarithmic phase at an optical density (OD) of 0.2 at 660 nm as the initial OD. The OD values, which reflect the turbidity of the cultures, were measured after appropriately diluting samples in the same medium to maintain a linear range of 0.1 to 0.6 OD (Ultrospec 500 pro, Amersham biosciences).” (line 77-86).

4.    Why did the authors use trehalose as the carbon source during the last phase of fed-batch culture? It must be discussed.

The choice of trehalose as a carbon source was guided by several factors. First, our HPLC analysis (Table S1) indicated that trehalose is the most abundant sugar present in the YP medium, likely derived from the yeast extract component. Additionally, thanks to the presence of trehalase genes, yeast readily uses trehalose and did not lead to catabolite repression. Given that the lack of ammonia production observed in YPD medium, which could be associated with the presence of glucose, we hypothesized that using trehalose would prevent the reuptake of ammonia from the medium, promoting continuous ammonia secretion as a result of amino acid uptake. We have further discussed this point in the discussion section by adding this paragraph:  “We selected trehalose as a carbon source because HPLC analysis (Table S1) showed it is the most abundant sugar in the YP medium, likely from the yeast extract component. Moreover, yeast readily utilizes trehalose without causing catabolite repression. Since ammonia production is hindered by glucose in YPD medium (Figure 1E), we hypothesized that trehalose would prevent ammonia reuptake and promote its continuous secretion from amino acid uptake”. (lines 367-372).

COMMENTS ON ENGLISH 
The English is understandable and good, but it could be improved. Some sentences could be wordy, and there are some cases where it would be advisable to split a long sentence into two short sentences to clarify the text. It is recommended that authors carefully review the English throughout the manuscript. 

We thank the reviewer for this comment. We have splitted long sentences throughout the text for clarification and revised the entire manuscript.

SPECIFIC COMMENTS

1. Line 53. Please consider the sentence "This study uses the vast biological..." to be the beginning of a new paragraph.
We thank the reviewer for the suggestion and have initiated a new paragraph.
2. Line 65-81. Please provide the manufacturer and model of the spectrophotometer used to measure OD.
We thank the reviewer for the suggestion. We have changed the paragraph with detailed information about the OD measure and spectrophotometer information (lines 81-84).
3.      Line 72. Was 0.2 OD the initial OD of the culture? If so, please state it clearly.
We have modified the paragraph detailing the initial OD statements (lines 79).
4.      Line 74. Please provide the volume and frequency of the sample collection.
We have provided sample volume and frequency as requested (lines 82-84).
5.      Line 79. Centrifugation speed never must be expressed in "rpm" because it changes according to the diameter of the rotor used. Please use "×g" rather than "rpm" and review it alongside the manuscript.
We thank the reviewer for their comment. We have updated the units for the centrifugal steps throughout the manuscript (lines 85, 92 and 104 ).
6.      Line 101. Was trehalose incorporated gradually or in a single addition? Please clarify.
We have clarified at line 114 by modifying this sentence: “When no changes in the optical density were observed, we started the third phase by adding 3% trehalose as a single addition”.
7.      Line 108-109. Please do not mention amino acids since this paragraph describes the determination of ammonia, trehalose, glycerol, and acetic acid.
We thank the reviewer and have removed the mention (line 122).
8.      Line 144-147. Was this screening made at the end of the 10 cycles of inoculation?
We have reviewed the protocol and introduced the following change: “This process was repeated a total of 10 times, during which evolved clones from different cycles were selected and tested.” (Lines 153-154).
9.      Line 201. Please provide these data as a supplementary table or figure. The journal's instruction for authors states that "data not shown" should be avoided.  
We thank the reviewer for clarifying the journal’s instructions. We have added trehalose consumption data at the time of inoculation and at the end of growth as supplementary material (Table S2). 
10.    Lines 225 and 227. Did the authors express the growth rate in units µ-1? What does it mean? Please review.
We have corrected the notation to h-1 (lines 249,251).

11.    Authors use periods or commas as decimal separators interchangeably throughout the manuscript. Please homogenize the use of the decimal separator.

We have homogenized commas and decimals separators throughout the manuscript.

Round 2

Reviewer 2 Report

Comments and Suggestions for Authors

the author revised the comments.

Author Response

We appreciate the reviewer’s feedback.

Reviewer 3 Report

Comments and Suggestions for Authors

Regarding growth measurement, I still have a serious concern. The effect that the optical configuration of a spectrophotometer has on optical density measurements has been well documented from several decades ago [Koch, A.L. (1970), Analytical Biochemistry, 38: 252-259; Koch, A.L. (1968), Journal of Theoretical Biology, 18: 133-156; Lawrence, J.V. and Maier, S. (1977), Applied and Environmental Microbiology, 482-484]. Then, instruments with different optical configurations will measure different optical densities for the same microbial suspension.

The practice of diluting a sample to measure the optical density in the linear range of the instrument and then multiplying by the dilution factor could be acceptable for monitoring and controlling a fermentation within a company that will not share or compare its data with any external agent. However, in my opinion, for a scientific publication, this is not acceptable because if another researcher tries to replicate the experiments and has a different instrument, she/he will get different readings, and those differences will be magnified when multiplying by the dilution factor. These differences will be more significant when the dilution factor is more prominent, as is the case with the manuscript's results under evaluation.

The correct practice is to build a standard curve using different cell densities (cells/mL or CFU/mL) and measure the optical density within the instrument's linear range. This standard curve allows the optical density to be read and transformed into cell density. Then, the dilution factor can multiply the cell density of the diluted sample. In this way, cell density measurement is now independent of the instrument. It is what I consider good laboratory practice.

Therefore, authors must construct a standard curve and report their cell density results in cells/mL or CFU/mL.

Minor comment: It is missing a comma before "and" [Line 121].

Author Response

Point-by-point response to Comments and Suggestions for Authors

Point 1. Regarding growth measurement, I still have a serious concern. The effect that the optical configuration of a spectrophotometer has on optical density measurements has been well documented from several decades ago [Koch, A.L. (1970), Analytical Biochemistry, 38: 252-259; Koch, A.L. (1968), Journal of Theoretical Biology, 18: 133-156; Lawrence, J.V. and Maier, S. (1977), Applied and Environmental Microbiology, 482-484]. Then, instruments with different optical configurations will measure different optical densities for the same microbial suspension.

The practice of diluting a sample to measure the optical density in the linear range of the instrument and then multiplying by the dilution factor could be acceptable for monitoring and controlling a fermentation within a company that will not share or compare its data with any external agent. However, in my opinion, for a scientific publication, this is not acceptable because if another researcher tries to replicate the experiments and has a different instrument, she/he will get different readings, and those differences will be magnified when multiplying by the dilution factor. These differences will be more significant when the dilution factor is more prominent, as is the case with the manuscript's results under evaluation.

The correct practice is to build a standard curve using different cell densities (cells/mL or CFU/mL) and measure the optical density within the instrument's linear range. This standard curve allows the optical density to be read and transformed into cell density. Then, the dilution factor can multiply the cell density of the diluted sample. In this way, cell density measurement is now independent of the instrument. It is what I consider good laboratory practice.

Therefore, authors must construct a standard curve and report their cell density results in cells/mL or CFU/mL.

We appreciate the reviewer’s feedback regarding this point. We agree that a correlation curve will make our data more straightforward and usable by many researchers. However, since our paper focuses on ammonia production in relation to overall biomass, we respectfully believe that the best way to express the data is to measure the biomass in terms of dry weight.  In our paper - and all papers of similar subject- absorbance is used as a proxy of cell mass (Ehrmann et al., Engineering Saccharomyces cerevisiae for fast vitamin-independent aerobic growth, (2024) Metabolic Engineering, 82; 201-215; Sreenivas et al., Evaluation of Pyrophosphate-Driven Proton Pumps in Saccharomyces cerevisiae under Stress Conditions. Microorganisms 2024, 12;625; Solomon, J.B., Lee, C.C., Liu, Y.A. et al. Ammonia synthesis via an engineered nitrogenase assembly pathway in Escherichia coli. Nat Catal 7, 1130–1141 (2024)). Therefore, we substituted OD with dry weight, an accurate measurement of cell mass that overcomes the inherent OD-related problems pointed out by the reviewer, providing a less ambiguous parameter than cell count (that is affected by the average size of the cells) or viable cell count (which can be misleading, as cell mortality may underestimate the biomass obtainable from a particular growth medium over time, in particular at late time points.

In light of these considerations, we have revised the optical density measurements reported in Figures 1A, 1B, 2A, 2B, and 3A, C, and D to reflect dry weight (g/L). We have established a correlation between optical density and dry weight using previously acquired data about the same kinetics shown in the manuscript. The regression analyses were performed for shake flask experiments for each strain (see Table S1 and Figure S1), while for data describing bioreactor experiments (Figure 3) we inserted the experimental values actually measured.

Additionally, we have added a new section in the Methods and Materials titled “2.3. Correlation between OD660 nm and dry weight” which describes the process in detail, along with the regression analysis information included in the supplementary materials. We have made minor modifications to paragraph 2.1 to incorporate information about the optical density acquisition, now included in the new paragraph 2.3. The ratio TN/OD has been converted to TN/DW, as indicated in paragraph 2.10, and Figures 1F and 2F have been updated accordingly. The statistical analysis for the TN/DW comparison in both Figures 1F and 2F has been repeated, revealing no significant differences compared to the previous ratio, though two more CEN.PK113-7D-derived clones were found to be statistically significant. We have also updated the entire manuscript to reflect the change from optical density measurements to dry weight.

We hope that these changes sufficiently satisfy the request.

Point 2. Minor comment: It is missing a comma before "and" [Line 121].

We have corrected the mistake.